# Depressive-like Behavior Is Accompanied by Prefrontal Cortical Innate Immune Fatigue and Dendritic Spine Losses after HIV-1 Tat and Morphine Exposure

**DOI:** 10.3390/v15030590

**Published:** 2023-02-21

**Authors:** Sara R. Nass, Yun K. Hahn, Michael Ohene-Nyako, Virginia D. McLane, M. Imad Damaj, Leroy R. Thacker, Pamela E. Knapp, Kurt F. Hauser

**Affiliations:** 1Department of Pharmacology and Toxicology, Medical College of Virginia (MCV) Campus, Virginia Commonwealth University, Richmond, VA 23298-0613, USA; 2Department of Anatomy and Neurobiology, Medical College of Virginia (MCV) Campus, Virginia Commonwealth University, Richmond, VA 23298-0709, USA; 3Department of Biostatistics, Medical College of Virginia (MCV) Campus, Virginia Commonwealth University, Richmond, VA 23219, USA; 4Institute for Drug and Alcohol Studies, Medical College of Virginia (MCV) Campus, Virginia Commonwealth University, Richmond, VA 23298-0059, USA

**Keywords:** opioid drug abuse, HIV associated neurocognitive disorders (HAND), depression, anhedonia, interleukin-10 (IL-10), C-C motif chemokine ligand 3 (CCL3), tumor necrosis factor α (TNFα), C-X-C motif chemokine ligand 1 (CXCL1), interleukin-6 (IL-6), C-C motif chemokine ligand 11 (CCL11)

## Abstract

Opioid use disorder (OUD) and HIV are comorbid epidemics that can increase depression. HIV and the viral protein Tat can directly induce neuronal injury within reward and emotionality brain circuitry, including the prefrontal cortex (PFC). Such damage involves both excitotoxic mechanisms and more indirect pathways through neuroinflammation, both of which can be worsened by opioid co-exposure. To assess whether excitotoxicity and/or neuroinflammation might drive depressive behaviors in persons infected with HIV (PWH) and those who use opioids, male mice were exposed to HIV-1 Tat for eight weeks, given escalating doses of morphine during the last two weeks, and assessed for depressive-like behavior. Tat expression decreased sucrose consumption and adaptability, whereas morphine administration increased chow consumption and exacerbated Tat-induced decreases in nesting and burrowing—activities associated with well-being. Across all treatment groups, depressive-like behavior correlated with increased proinflammatory cytokines in the PFC. Nevertheless, supporting the theory that innate immune responses adapt to chronic Tat exposure, most proinflammatory cytokines were unaffected by Tat or morphine. Further, Tat increased PFC levels of the anti-inflammatory cytokine IL-10, which were exacerbated by morphine administration. Tat, but not morphine, decreased dendritic spine density on layer V pyramidal neurons in the anterior cingulate. Together, our findings suggest that HIV-1 Tat and morphine differentially induce depressive-like behaviors associated with increased neuroinflammation, synaptic losses, and immune fatigue within the PFC.

## 1. Introduction

Opioid use disorder (OUD) is an escalating problem not only in the United States but worldwide [1] as the rates of prescription and illicit (including fentanyl-contaminated street drugs) opioid use and resultant overdoses increase. Not only can opioid misuse increase the risk of infection with bloodborne diseases such as HIV-1 [2], but opioids can worsen the pathogenesis of neuroHIV and HIV-associated neurocognitive disorders (HAND) [3,4,5,6,7]. OUD, HIV, and mood disorders are overlapping comorbidities [8,9] that can complicate treatment and lead to decreased medication adherence [9]. Importantly, the neuropathogenic mechanisms thought to underly HAND, and opioid exacerbation of neuroHIV (e.g., neuroinflammation [10,11,12] and synaptodendritic injury [13,14,15]) are also implicated in major depressive disorder (MDD).

Up to a third of patients with substance use disorder (SUD) have comorbid mental health disorders (e.g., depression) [16,17]. Anhedonia—the lack of pleasure in previously enjoyable things—is frequently seen in patients with OUD and correlates with opioid craving and misuse [18,19]. Further, the opioid receptor antagonist naltrexone decreases self-reported depression in heroin-dependent individuals [20]. Individuals with OUD also exhibit increased expression of extracellular matrix transcripts in the PFC [21,22] that are implicated in dendritic spine formation and synaptic remodeling [23]. Although opioids tend to suppress peripheral immunity [24], the CNS appears to respond differently. Individuals with OUD exhibit increased proinflammatory cytokines and inflammatory markers in the PFC (e.g., CD40, interleukin-33 (IL-33), IL-17 receptor B (IL-17RB) and increased nucleotide-binding oligomerization domain containing 1 (NOD1) pattern recognition receptors [22], which are repeatedly aggravated by opioid withdrawal [25,26,27] during the withdrawal and relapse cycle of addiction [28].

Neuropsychiatric disorders are a major problem in persons infected with HIV (PWH), with the prevalence of self-reported depressive symptoms estimated to be around 40% [29,30,31], and two-fold higher rates of MDD compared to HIV-negative controls are reported in clinical studies [31]. In PWH, depression most likely stems from (i) biological factors, including viral protein-induced central nervous system (CNS) excitotoxicity and inflammation, and (ii) psychosocial factors, such as lack of support, stigma, and chronic disease burden [29,30]. In multiple cohorts of PWH, depressed individuals have higher plasma proinflammatory C-X-C motif chemokine ligand 10 (CXCL10), IL-12, granulocyte colony-stimulating factor (G-CSF), and C-reactive protein (CRP) levels than non-depressed individuals [32,33]. Moreover, PWH with depressive symptoms (as measured by Beck depression inventory II) had worse neurocognitive outcomes than non-depressed PWH over three years, which coincided with increasing plasma CRP levels [34]. There is increased microgliosis—assessed by the percentage of area with CD68 immunoreactive cells in the hippocampus, frontal cortex, and basal ganglia of individuals with neuroHIV and OUD despite the use of combination antiretroviral therapy (cART) [35,36]. Further, synaptic injury, which is seen in patients with MDD [13,15], can occur in virally suppressed PWH [37]. Collectively, heightened inflammation and synaptic stress are likely to contribute to the higher prevalence of depression in neuroHIV and OUD comorbidity.

In the CNS, the HIV-1 protein Tat can be detected in a subset of PWH even in the presence of antiretroviral therapies [38,39]. Tat induces morphological and functional deficits in neurons through direct excitotoxic mechanisms [40,41,42] and indirectly through neuroinflammatory mediators [43,44,45,46], which can be worsened by opioid co-exposure [40,45,47,48]. Tat transgenic (tg) mice mimic many of the cognitive and emotional behavioral deficits and region-specific neuroinflammation and pathogenesis seen in PWH with [27,47,49,50,51] or without [52,53,54,55,56,57,58,59,60] opioid co-exposure. Previous research suggests HIV-Tat induces depressive-like behavior in male and proestrus female mice [49,58,61,62], whereas decreased adaptability correlates with PFC proinflammatory cytokines and chemokines expression in Tat(+) mice [59]. However, the only study to look at opioid coadministration found that acute oxycodone administered 20 min before testing increased mobility in the tail suspension test in female mice irrespective of Tat exposure or estrus phase—presumably due to opioid-induced hyper locomotor confounds [49]. Therefore, the role of comorbid opioids, particularly chronic administration, in HIV-Tat-induced depressive-like behavior and the association with neuroinflammation and synaptodendritic injury remains unclear.

The goal of the present study was to quantify behavioral deficits associated with MDD in response to HIV-Tat and repeated morphine exposure in transgenic mice and to examine the extent that depressive-like behavior coincides with neuroinflammation and synaptodendritic pathology in the PFC. Our hypothesis is that sustained HIV-1 Tat and morphine exposure will interact to induce depressive-like behaviors and decreased adaptability, and this will coincide with perturbations in cytokine production and dendritic spine density within the PFC.

## 2. Materials and Methods

Virginia Commonwealth University Animal Care and Use Committee approved all procedures using mice. All experiments were conducted in accordance with the National Institutes of Health (NIH Publication No. 85–23) ethical guidelines.

### 2.1. Subjects and Doxycycline Treatment

Adult male doxycycline (DOX)-inducible HIV-1_IIIB_ Tat_1–86_ transgenic (tg) mice (3–5-months-old) mice (*n* = 79) were used in all studies. HIV-Tat is GFAP-driven under the control of a reverse tetracycline-controlled transactivating (rtTA) transcription factor. Tat(−) (control) littermates expressing the rtTA transcription factor but not the *tat* transgene were used as controls [63]. HIV-1 Tat-tg mice were bred and maintained in 3–4 per cage in a temperature- and humidity-controlled facility, with ad libitum access to food and water, on a 12:12 light: dark cycle in the AAALAC-accredited vivarium of Virginia Commonwealth University as previously described [59,63,64]. To induce the expression of HIV-1 Tat(+), mice were fed a DOX-supplemented chow (6 g/kg, Harlan Laboratories, Madison, WI) for the entire experimental duration (8 weeks). Tat(−) mice were also fed DOX-chow to control for off-target effects of DOX.

### 2.2. Drug Treatment

Starting at day 43 (week 6) of DOX administration, mice were administered escalating, subcutaneous (s.c.) doses of morphine sulfate (Drug Supply Program, National Institute on Drug Abuse, Bethesda, MD) dissolved in sterile saline (10–40 mg/kg administered twice daily (b.i.d.), increasing by 10 mg/kg at 2-day intervals) or saline until the end of the experiment (2 weeks total) as previously described (Figure 1 [51,65]). All solutions were warmed to room temperature before being administered at a volume of 10 µL/g body weight.

### 2.3. Behavioral Assays

Mice were assessed for anhedonia, well-being, and acute environmental stressor adaptability in multiple assays across the domain of depressive-like behavior. In order to minimize animal numbers, mice were divided into two behavioral cohorts and run through one of the two following batteries of behavioral tests: (cohort 1, *n* = 39) sucrose preference, novelty-induced sucrose hypophagia, nestlet shredding, and burrowing or (cohort 2, *n* = 40) forced swim test and novelty-suppressed feeding (Figure 1). Testing was conducted in ascending order of presumed stress to reduce carry-over effects. In order to control for opioid-induced hyperlocomotion, mice received morphine or saline and were habituated to the testing room for at least 4 h prior to behavioral testing [66,67]. At the conclusion of behavioral testing, rapid cervical dislocation was used to humanely euthanize the mice, brain tissues were collected, and tissues were randomly assigned for either (i) multiplex analysis and the PFC was dissected, snap frozen in liquid nitrogen, and stored at −80 °C or (ii) whole forebrain fixation to be impregnated using a Rapid Golgi procedure in the PFC (Figure 1). Experimenters were blinded to treatment conditions during all the experiments.

#### 2.3.1. Two-Bottle Choice Sucrose Preference

Cohort 1 mice were assessed for anhedonia-like behavior in two different assays that measured sucrose consumption in a home cage versus a novel environment. Cohort 1 mice were single-housed for 1 week before behavioral testing to acclimate and remained single-housed for the remainder of the experiment (3 weeks total; Figure 1). Starting on the 1st day of morphine or saline administration (day 43), singly housed mice were provided with a bottle of water and a bottle of 2% sucrose solution with ad libitum access to food in their home cages, as previously described [68,69]. Both bottles were weighed at the start of the experiment and every other day thereafter for 7 days. The bottle position was switched daily. After the two-bottle choice assay was completed, the sucrose bottles were removed, and mice were allowed to adjust for 7 days before the remaining behavioral assays were conducted. The percentage of sucrose preference for every 2 days was calculated by dividing the amount of sucrose liquid consumed by the total amount of all liquid consumed and multiplying by 100.

#### 2.3.2. Novelty-Induced Sucrose Hypophagia

Sucrose consumption in a novel environment was also used to assess anhedonia-like behavior using the novelty-induced sucrose hypophagia test with modifications [59,70]. The mice in cohort 1 were previously exposed to 2% sucrose in the two-bottle choice test and had ad libitum access to food throughout the duration of the experiment. On day 55 of Tat exposure, mice received morphine or saline, and 4 h later, were allowed to explore an open-field chamber (40 × 40 × 35 cm; Stoelting Co., Wood Dale, IL, USA) with a 60 mm-diameter Petri dish containing 2% sucrose solution placed in the center of the chamber. Movements of the mice were digitally recorded by ANY-maze version 6.35 software (Stoelting Co., Wood Dale, IL, USA), and an experimenter blinded to the treatment conditions coded the time spent consuming the sucrose solution using the ANY-maze software (Stoelting Co.) as a measure of anhedonia.

#### 2.3.3. Burrowing

Burrowing is a common innate behavior conserved in laboratory mice that appears to be related to shelter maintenance and defense [71]. Some researchers have suggested burrowing may represent activities of daily living [72] in humans that are often diminished with depression [73,74]. Briefly, at the end of three weeks of the single housing, a dry water bottle filled with pre-weighed DOX food pellets (~225 g) was placed in the back corner of the cage of each cohort 1 mouse, which had continuous ad libitum access to food. Mice were allowed to freely explore for 2 h, and the percentage of food pellets removed from the burrow was calculated by subtracting the weight of the remaining food pellets from the starting weight of the food pellets as a proxy measure of burrowing behavior [75,76].

#### 2.3.4. Nesting

The nestlet shredding/nest building test is often used in conjunction with the burrowing test to assess mouse well-being as a proxy measure of activities of daily living [72,75,77] and was performed as previously described with modifications [75,76]. Nestlet material was removed from each cage of cohort 1 mice and replaced with 1 pre-weighed pressed cotton nestlet (5 × 5 × 0.5 cm) 30 min before the dark cycle on day 55 of Tat exposure. The next morning, 14 h later, intact pieces of nestlet (i.e., all pieces that were still in pressed form) were removed and dried overnight before being weighed again. The percentage of the nestlet shredded was calculated by subtracting the weight of the remaining intact nestlet (after shredding) from the starting nestlet weight as a proxy measure of nest building [59].

#### 2.3.5. Forced Swim Test (FST)

Due to the possible confound of single-housing, a separate cohort of group-housed mice were also tested in the FST and novelty-suppressed feeding assay. Traditionally, the FST has been used as an acute antidepressant screen and model of behavioral despair [78,79], but the validity of these interpretations has been questioned [80,81,82]. Instead, a growing body of research suggests that a more accurate interpretation is that the FST measures coping behavior and adaptability in response to an acute inescapable stressor [83,84,85]. The FST was performed as previously described on day 55 of Tat exposure [59,69]. Briefly, mice were acclimated to a 2 L beaker of warm water (~25 °C) for 2 min. Immediately following the acclimation period, the time spent either swimming or climbing (i.e., struggling) was recorded for each mouse for 4 min (6 min total) as a measure of adaptive coping behavior.

#### 2.3.6. Novelty-Suppressed Feeding

Although the novelty-suppressed feeding test is often used to assess anxiety-like behavior, unlike the FST, it demonstrates face validity for the long-term administration of traditional antidepressant drugs [86,87]. Mice were assessed for novelty-suppressed feeding as previously described [59]. On day 55 of Tat exposure, mice in cohort 2 were food-deprived overnight for 14 h. On day 56, 4 h after morphine (or saline administration), a 60 mm-Petri dish filled with 6–8 DOX food pellets was placed in the center of an open-field chamber (40 × 40 × 35 cm; Stoelting Co.) that mice explored for 10 min. ANY-maze video recording cameras and software (Stoelting Co.) were used to record the amount of time the mice spent eating the food pellets as a measure of depressive-like behavior.

### 2.4. Cytokine Assessment

Following nestlet shredding/nest building (cohort 1) or novelty-suppressed feeding (cohort 2) assessments, mice were humanely euthanized via rapid cervical dislocation. After random assignment, either the whole PFC was grossly dissected, flash frozen in liquid nitrogen, and stored at −80 °C until used for multiplex analysis or the whole forebrain was dissected and impregnated using a Golgi procedure (see below). PFC chemokine/cytokine levels (i.e., CCL2, CCL3, CCL4, CCL5, CCL11, CXCL1, G-CSF, GM-CSF, TNFα, IFN-γ, IL-2, IL-3, IL-6, IL-9, IL-12p40, IL-12p70, IL-17A, IL-4, IL-5, IL-10, IL-13) were assessed using the Bio-Plex, Pro Mouse Cytokine 23-plex assay kit (cat. no. M60009RDPD Bio-Rad Laboratories, Inc., Hercules, CA, USA) per the manufacturer’s instructions and assayed using the Bio-Plex^®^ 200 System (Bio-Rad Laboratories) as previously described [27,59]. The samples were fitted to the respective five-parameter logistic standard curves using the Bio-Plex Manager 4.0 software (Bio-Rad, Hercules, CA, USA). All sample values fell below the highest standard value on each respective standard curve, and no values were outside the detection limits of the 23-plex assay kit.

### 2.5. Spine Density Assessment

The FD rapid GolgiStain^TM^ kit was used for Golgi impregnation according to the manufacturer’s directions (FD Neurotechnologies, Columbia, MD, USA) as previously described [59]. Spines were counted on clearly visualized dendritic segments, parallel to the plane of the section, from neurons that were fully impregnated, using a Zeiss Axio Examiner D1 microscope (Zeiss, Oberkochen, Germany) and reported as the number of spines per 10 µm length of dendrite. The Mouse Brain in Stereotaxic Coordinates (Franklin and Paxinos, 2008) atlas was used to differentiate regions within the PFC. Spines were counted on the apical, oblique, and basal dendrites of layer 5 pyramidal neurons in the ACC [59,88].

### 2.6. Statistical Analyses

Data were analyzed by analysis of variance (ANOVA) using Prism version 9.4.1 (GraphPad Software, Inc., San Diego, CA, USA). Behavioral and PFC cytokine/chemokine expression data were analyzed by two-way ANOVA (genotype × drug treatment). Sucrose preference (genotype × drug treatment × time) and dendritic spine density (genotype × drug treatment × dendrite type) data were analyzed using repeated measures and three-way ANOVA. Planned pairwise comparisons were performed to delineate interactions (or lack thereof) between HIV-Tat and morphine using Tukey’s HSD to correct the experiment-wise error rate. Partial eta squared (η^2^) and Cohen’s *d* was used to denote effect size. Pearson’s correlation analyses were performed to assess the relationship between cytokine/chemokine levels within the PFC and behavioral outcomes. False discovery rate (FDR) corrections were performed to adjust for multiple correlation analyses. All data are presented as the mean ± the S.E.M. Differences were considered statistically significant if *p* < 0.05.

## 3. Results

### 3.1. Effects of HIV-1 Tat and Morphine on Depressive-like Behavior

After DOX administration for 8 weeks and escalated morphine (10–40 mg/kg, s.c., b.i.d.) or saline for 2 weeks, Tat(+) and Tat(–) mice were assessed for either (1) sucrose preference and behaviors associated with well-being, or (2) novelty-suppressed feeding and forced swim test behavior.

#### 3.1.1. HIV-1 Tat and Morphine Differentially Decreased Anhedonia-like Behavior

At the onset of morphine administration, single-housed mice were assessed for sucrose preference in the two-bottle choice assay for 1 week. There was an interaction between Tat exposure and the day of sucrose administration [*F*_(3,140)_ = 3.42, *p* < 0.05; η^2^ = 0.06; Figure 2A]. Planned comparisons revealed that Tat(+) and saline-treated mice exhibited significantly lower sucrose preference than Tat(–) and saline-treated mice on the 1st day of testing (*p* < 0.05; *d* = −0.96) but significantly more sucrose than Tat(–) and morphine-treated mice on the 5th day of testing (*p* < 0.05; *d* = 1.03). There was a main effect of morphine [*F*_(1,140)_ = 5.42, *p* < 0.05; η^2^ = 0.04; Figure 2A], but not Tat (*p* = 0.91), to decrease sucrose preference across all 7 testing days.

Mice were allowed to adapt to the sucrose bottle removal for one week and then, at the end of the morphine regimen, were tested in the novelty-induced sucrose hypophagia test. Main effects indicated that Tat [*F*_(1,35)_ = 20.55, *p* < 0.0001; η^2^ = 0.36; Figure 2B; *F*_(1,35)_ = 13.14, *p* < 0.001; η^2^ = 0.26; Figure 2C], but not morphine (*p* = 0.28; *p* = 0.38), increased the latency to approach and the amount of time spent consuming the sucrose solution in a novel environment, respectively. There was also a significant interaction between Tat and morphine in time spent consuming the sucrose solution in a novel environment [*F*_(1,35)_ = 4.64, *p* < 0.05; η^2^ = 0.09; Figure 2C]. Tukey’s HSD test indicated that Tat(+) mice administered saline (*p* < 0.01; *d* = −1.54) or morphine (*p* < 0.05; *d* = −1.15) spent significantly less time drinking the sucrose solution compared to Tat(–) saline administered mice. Tat also decreased [*F*_(1,35)_ = 19.71, *p* < 0.0001; η^2^ = 0.29; Figure 2D], whereas morphine increased [*F*_(1,35)_ = 8.47, *p* < 0.01; η^2^ = 0.12; Figure 2D] mobility in the novelty-induced hypophagia test. An interaction between Tat and morphine [*F*_(1,35)_ = 5.5, *p* < 0.05; η^2^ = 0.08; Figure 2D] and pairwise comparisons suggest these effects were mainly driven by Tat(–) mice administered morphine spending significantly more time mobile than other treatment groups (*p* < 0.01; *d* ≥ 1.85). However, there was no effect of Tat (*p* = 0.68; Figure 2E) or morphine (*p* = 0.22; Figure 2E) on rearing behavior. Overall, morphine decreased home cage sucrose preference, and Tat decreased the time mice consumed sucrose in a novel environment, suggesting a context-specific increase in anhedonia.

#### 3.1.2. HIV-1 Tat and Morphine Differentially Decreased Innate Behaviors of Well-Being

Cohort 1 mice were also assessed for innate home cage nestlet shredding and burrowing behaviors as proxy measures of well-being. Main effects showed that morphine [*F*_(1,35)_ = 10.03, *p* < 0.01; η^2^ = 0.21; Figure 3A], but not Tat (*p* = 0.13), decreased the percentage of food burrowed. Planned comparisons suggest that Tat and morphine co-exposure are driving this effect resulting in burrowing behavior that is significantly less than in Tat(–) and saline-treated mice (*p* < 0.01; *d* = 1.38). In contrast, regardless of drug administration (*p* = 0.10), Tat decreased the amount of nestlet shredded overnight [*F*_(1,35)_ = 8.79, *p* < 0.01; η^2^ = 0.19; Figure 3B]. However, planned comparisons indicate that Tat(+) and morphine-treated mice shredded significantly less nestlet material than Tat(–) and saline-exposed mice (*p* < 0.01; *d* = −1.46).

#### 3.1.3. HIV-1 Tat, but Not Morphine Decreased Adaptation to an Acute Stressor

At the end of 2 weeks of ramping morphine (or saline) treatments, mice in cohort 2 were assessed for the ability to cope with a non-escapable stressor in the forced swim test. The test revealed that Tat increased mobility [*F*_(1,36)_ = 10.95, *p* < 0.001; η^2^ = 0.23; Figure 3C], suggesting a decrease in coping ability because Tat-exposed mice were not switching to a more appropriate passive coping strategy to deal with this inescapable stressor. Planned comparisons revealed that Tat(+) saline-treated mice were significantly more mobile than Tat(–) saline-treated mice (*p* < 0.05, *d* = 1.42). Morphine did not alter mobility (*p* = 0.84) in the FST.

#### 3.1.4. Morphine, but Not HIV-1 Tat Increased Novelty-Suppressed Feeding

Finally, we assessed depressive-like behavior in cohort 2 mice in the novelty-suppressed feeding test after overnight food deprivation. There was no effect of morphine (*p* = 0.85) or Tat (*p* = 0.84) on latency to approach DOX food pellets in the center of the chamber (Figure 4A). However, the main effects revealed that morphine [*F*_(1,36)_ = 14.23, *p* < 0.001; η^2^ = 0.27; Figure 4B], but not Tat (*p* = 0.21), increased time spent feeding in the novel environment. Planned comparisons suggest that mice co-exposed to Tat and morphine spent more time feeding than Tat(–) (*p* < 0.01; *d* = 1.49) or Tat(+) (*p* < 0.05; *d* = 1.47) mice administered saline. Further, morphine [*F*_(1,36)_ = 17.02, *p* < 0.001; η^2^ = 0.31; Figure 4C], but not Tat (*p* = 0.33), had a main effect of decreasing locomotion, whereas neither altered rearing behavior (Tat, *p* = 0.89; morphine, *p* = 0.98; Figure 4D). Planned comparisons suggest that mice exposed to Tat and saline were more mobile than Tat(–) (*p* < 0.01; *d* = 1.57) or Tat(+) (*p* < 0.01; *d* = 1.64) mice that were given morphine.

### 3.2. PFC Cytokines and Chemokines Are Minimally Altered by Long-Term HIV-1 Tat and Repeated Morphine Exposure

After the conclusion of behavioral testing, chemokine and cytokine levels in the PFC were assessed via multiplex assay. Statistical parameters and outcomes of all tested chemokines and cytokines (*n* = 23) are reported in Table 1. For clarity, only chemokines and cytokines that displayed significant (*p* < 0.05) or trending (*p* < 0.10) differences in response to Tat or morphine exposure are graphically displayed (Figure 5). The means and standard errors of all other chemokines and cytokines are reported in Table 2. Tat induction for 8 weeks significantly increased proinflammatory chemokine CCL3 [*p* < 0.05; η^2^ = 0.18; Table 1, Figure 5A] and anti-inflammatory cytokine IL-10 [*p* < 0.05; η^2^ = 0.18; Table 1, Figure 5C] levels. Planned comparisons revealed that Tat(+) and morphine-treated mice expressed significantly higher PFC IL-10 levels than Tat(–) mice exposed to saline (*p* < 0.05; *d* = −1.73). Tat also tended to decrease proinflammatory IFN-γ [*p* = 0.053; η^2^ = 0.17; Table 1, Figure 5D] and IL-6 [*p* = 0.09; η^2^ = 0.13; Table 1, Figure 5F] levels, whereas Tat tended to increase the anti-inflammatory cytokine IL-4 [*p* = 0.07; η^2^ = 0.15; Table 1, Figure 5E]. In contrast, 2 weeks of escalating morphine exposure decreased levels of the proinflammatory cytokine TNFα [*p* < 0.05; η^2^ = 0.22; Table 1, Figure 5B].

#### PFC Chemokines and Cytokines Differentially Correlated with Depressive-like Behaviors in HIV-1 Tat and Morphine Exposed Mice

Correlation analyses were used to investigate the relationship between PFC chemokine and cytokine levels and depressive-like behavior in Tat- and morphine-treated mice. Correlation coefficient *z*-tests were performed with hypothesized correlations equal to 0. Correlations that contained both moderate coefficients (i.e., *r* > 0.56) and significant uncorrected *p*-values (i.e., *p* ≤ 0.05) are graphically displayed (Figure 6). Statistical parameters are presented in Appendix A, including correlation coefficients, *z*-values, and *p*-values (uncorrected and FDR corrected) for all correlations between chemokines/cytokines and behavioral outcomes that are not graphically displaced in Figure 6. Across all treatment groups, the latency to approach the sucrose solution directly correlated with increases in CXCL1 levels [*r*_(11)_ = 0.56, *p* < 0.05, FDR corrected *p* = 0.47; Figure 6A] in the novelty-induced hypophagia test. Time spent consuming sucrose in the novelty-induced hypophagia test inversely correlated with IL-10 levels [*r*_(11)_ = −0.61, *p* < 0.05, FDR corrected *p* = 0.61; Figure 6B]. Rearing time in the novelty-induced hypophagia test [*r*_(11)_ = −0.69, *p* < 0.01, FDR corrected *p* = 0.18; Figure 6C] and burrowing behavior [*r*_(11)_ = −0.70, *p* < 0.01, FDR corrected *p* = 0.15; Figure 6D] negatively correlated with IL-6 levels. Further, in the novelty-suppressed feeding test, mobility [*r*_(9)_ = −0.72, *p* < 0.01, FDR corrected *p* = 0.22; Figure 6E] and rearing [*r*_(9)_ = −0.82, *p* < 0.01, FDR corrected *p* = 0.03; Figure 6F] inversely correlated with increasing CCL11 levels.

### 3.3. HIV-1 Tat Decreased Anterior Cingulate Cortex (ACC) Dendritic Spine Density on Pyramidal Neurons

The ACC is important for emotional regulation, motivation, and adaptability [89,90,91]. After behavioral testing, dendritic spine density on layer V pyramidal neurons within the ACC region of the PFC was assessed in Golgi-impregnated brain sections. A main effect of Tat [*F*_(1,69)_ = 4.72, *p* < 0.05; η^2^ = 0.06; Figure 7], but not morphine (*p* = 0.99), indicated that Tat decreased the dendritic spine density across all pyramidal neuron dendritic types (i.e., apical, oblique, and basal dendrites).

## 4. Discussion

In the current study, Tat and morphine-induced depressive-like behavior and decreased innate and coping behaviors. The changes in behavior were associated with net reductions in proinflammatory cytokine levels in the PFC, indicative of innate immune fatigue. Tat also decreased spine density on pyramidal neurons in layer V of the ACC. Mice exposed to Tat for eight weeks had a longer latency to approach and a decrease in consumption of sucrose in the novelty-induced hypophagia test, irrespective of morphine treatment, suggesting anhedonia-like behavior. This aligns with a previous report in a similar model with higher levels of *tat* gene expression [92], which demonstrated decreases in saccharin consumption after one week of Tat exposure [58]. Furthermore, in the same model, Tat induction over one week increased intracranial self-stimulation (ICSS) response thresholds, indicating more current was required to maintain similar reward responding during baseline testing [93]. However, we previously found that Tat exposure increased the time spent consuming sugar cereal after two weeks but not four to eight weeks [59], suggesting alterations in behavioral pathology depending on the duration of exposure. In support, Tat decreases in sucrose preference after six weeks of exposure were reversed at five days of sucrose bottle presentation in saline-treated mice. However, surprisingly, control Tat(–) saline-treated mice exhibited decreased sucrose preference over time, which is normally stable—potentially due to injection stress [94,95]. Regardless, decreases in compensatory inflammatory cytokines and normalization of aberrant neuronal firing rates and behavior become more apparent with a longer duration of Tat exposure [27,55,59,96,97].

Morphine also decreased sucrose preference in the two-bottle choice test and exacerbated Tat-induced decreases in time spent consuming sucrose in the novelty-induced hypophagia test. These data align with reports indicating increases in anhedonia in OUD [18] and suggest the heightened anhedonia could be exacerbated in PWH. However, these data are also somewhat surprising given that acute opioid administration increases sugar intake in rats [98,99] or the preference for sweets in people with OUD [100], whereas the opioid receptor antagonists naloxone and naltrexone decrease palatable food preference [101]. However, both acute morphine during short sucrose presentation sessions [102] and repeated morphine exposure for two weeks [103] decreased sucrose intake in rats, suggesting alterations in the timing of sucrose presentation and opioid administration might account for these differences. Moreover, in most preclinical studies, μ opioid receptor (MOR) agonists and antagonists are intracranially injected directly into brain areas associated with reward that are also often implicated in feeding behavior (e.g., the nucleus accumbens and hypothalamus) [100]. Differences between the present study (decreased sucrose preference) and the OUD clinical population (increased consumption of sugary foods) might also be because liquid carbohydrates are less satiating than solid carbohydrates [104]. Indeed, people with OUD rate sucrose solutions less pleasant (lower concentrations) or similarly pleasant (higher concentrations) when compared to individuals without OUD [105].

Despite the decrease in sucrose preference, morphine, particularly in the Tat(+) mice, increased consumption of the DOX chow after overnight food deprivation in the novelty-suppressed feeding test. This expands on our previous report indicating that Tat exposure increased feeding in the novelty-suppressed feeding test [59]. Although heroin misuse is often associated with a lack of food consumption and an underweight basal metabolic index (BMI), many people with OUD experience a comorbid binge-eating disorder and food cravings [106]. Further, OUD patients often experience weight gain and increased sugar cravings while under opioid substitution therapy or in treatment centers [105,107], suggesting a role of food scarcity in the eating habits of people with OUD not undergoing therapy. In rodents, stimulation of MOR in the PFC promotes feeding behavior [108], in which rapid behavioral switching between short periods of feeding and exploratory locomotor activity is particularly evident [109]. This dysregulated feeding pattern might explain the similar decrease in body weight after chronic administration of opioids in rodents, whereas overall feeding is decreased during the first few days of opioid administration, followed by an inverted time-of-day feeding schedule [110].

Quality of life—including the ability to perform activities associated with daily living—appears to be preferentially decreased in PWH using opioids than in either population alone [111,112]. To assess well-being, we measured innate mouse behaviors that are proposed to model activities of daily living [72,75,77]. Morphine exacerbated the Tat-induced decreases in overnight nestlet shredding—a proxy measure of nesting behavior. Morphine also exacerbates the reduction of mouse nesting behavior in the 2,4,6-trinitrobenzene sulfonic acid (TNBS) model of experimental colitis [113]. Although nestlet shredding did not correlate with PFC cytokine levels, Tat and morphine induce inflammation and neuronal dysfunction in the enteric nervous system [114,115,116], which may contribute to the reduced nesting behavior in the present study. Interestingly, previous research found no difference in nestlet shredding in Tat(+) mice in non-home test cages for 2 h [59,117], indicating timing and environmental factors also affect Tat-induced reductions in nest-building behavior. The morphine-induced decrease of burrowing was also exacerbated in Tat(+) mice, suggesting that Tat and morphine in combination are particularly detrimental to well-being. However, a Tat-induced increase in FST mobility was evident in saline-treated mice, suggesting that morphine does not contribute to Tat-induced decreased adaptability in response to the acute swim stressor. This coincides with our previous data in Tat-tg mice not exposed to morphine [59].

Across all treatment groups, we saw correlations between increased proinflammatory cytokines and chemokines and increased depressive-like behavior or decreased mobility. For example, in the present study, the proinflammatory chemokine CXCL1 positively correlated with increased latency to seek sucrose in the novelty-induced hypophagia test. We previously found that in Tat(+) mice, CXCL1 positively correlated with immobility in the FST. Further, in the chronic, unpredictable, mild stress model of depression, increased levels of CXCL1 mRNA and protein in the hippocampus [118] and protein in cerebrospinal fluid [119], respectively, and increased expression of its cognate receptor CXCR2 [119], coincided with decreased sucrose preference that was reversed by the selective serotonin reuptake inhibitor (SSRI) fluoxetine [119]. Increased PFC levels of the proinflammatory chemokine CCL11 also correlated with decreased rearing and mobility in the novelty-suppressed feeding test. This aligns with previous data in patients with schizophrenia [120] and the 1-methyl-4-phenyl-1,2,3,6-tetrahydropyridine (MPTP) acute mouse model of Parkinson’s disease [121]. Overall, these correlations suggest depressive-like behavior is associated with increased levels of specific proinflammatory cytokines in key brain regions, supporting the neuroinflammatory theory of depression.

Previous research has shown increased astrogliosis and microglial activation in the CNS of PWH that misuse opioids even in the presence of cART [35,36,122], and increased proinflammatory cytokine levels and microglial activation in patients with MDD, especially those with suicidality [123]. In the present study, proinflammatory chemokine CCL3 levels were increased in the PFC of Tat(+) mice. Similarly, the presence of a CCL3 rs1130371 polymorphism and depression predict increased HAND in PWH [124]. Although we previously did not see Tat-induced alterations in CCL3 expression in the PFC after 48 h, 2 weeks, or eight weeks of exposure [59], the results in the present study appear to be driven by morphine. CCL3 is a cognate ligand for the CCR5 receptor [125], the co-receptor for R5 tropic HIV, which is the predominant strain found in the CNS [126,127]. CCR5 expression by astroglia appears to mediate opioid exacerbation of HIV-Tat neurotoxicity in vitro [128].

Innate immune fatigue or “tolerance” within CNS resident microglia [129,130,131], or compensatory immunosuppression [132], is seen after repeated or sustained inflammatory insult and compelling evidence suggests it could be implicated in neurodegenerative disorders [133,134,135]. The anti-inflammatory cytokine IL-10 and the transforming growth factor-β (TGF-β) are essential to immune regulation, tissue protection, and resolution of inflammation [133]. However, the immunosuppression associated with repeated or sustained insults can disrupt homeostasis and contribute to decreased pathogen and pathological cell/protein clearance [133]. For example, after repeated systemic administration of lipopolysaccharides (LPS), proinflammatory cytokine levels decrease, whereas anti-inflammatory IL-10 levels increase within the brain [134]. Further, in the amyloid precursor protein/presenilin 1 (AAP/PS1) model of Alzheimer’s disease, IL-10 knockout mice exhibit decreased amyloid-β deposits in their hippocampus and cortex [136]. Innate immune fatigue within the striatum, hippocampus, and spinal cord are seen after prolonged Tat (up to 3 months) exposure [27,97,137]. Expanding on our previous data showing that 48 h–8 weeks of Tat exposure increases PFC expression of the anti-inflammatory cytokine IL-10 [59], morphine exacerbated the Tat-induced increase of IL-10 expression in the PFC. Although the results are mixed, IL-10 also appears to be elevated in patients with MDD [138,139]. Further, hippocampal microglial IL-10 is increased in mice that are stressed via exposure to a learned-helplessness paradigm [140], suggesting sustained inflammation may not be necessary for depressive-like behavior after the immune system is activated. Indeed, even though depressive symptomology resolves with cessation of IFNα immunotherapy in individuals with hepatitis C, patients that experienced depression during treatment have a greater chance of developing recurrent depressive symptoms [141]. This may explain why Tat tended to decrease PFC IL-6 and IFN-γ levels, which are implicated in depression [142,143], even though Tat(+) and morphine co-exposed mice exhibited greater depressive-like behavior overall than control mice. Despite findings of innate immune suppression following prolonged Tat exposure in the PFC, the reductions in cytokines are likely preceded by pronounced increases in response to acute Tat ± morphine some proinflammatory cytokines correlated with increased depressive-like behavior exposure as seen in other CNS regions in vivo [47] and in vitro [144,145]. Further, IL-6 and IFN-γ are suppressed by the SOCS3 (suppressor of cytokine signaling 3) negative feedback loop, while IL-10 is not suppressed [146] and can inhibit IFN-γ by itself [147]. In PWH, increased serum IL-10 levels inversely correlate with IFN-γ levels [148]. These data suggest the anti-inflammatory counterbalance to sustained or repeated insult may lead to immune dysregulation and contribute to neuropathology and behavioral dysfunction.

An initial release of proinflammatory cytokines, which has been seen in vivo in the striatum in our Tat(+) mice 24 h after exposure [27] or in the whole brain 4 h after a single intracerebroventricular (icv) Tat injection [61], may explain why despite innate immune fatigue following sustained Tat exposure. Proinflammatory cytokines may induce depression by activating indoleamine 2,3 dioxygenase (IDO), which restricts serotonin synthesis by converting the serotonin precursor tryptophan into kynurenine [149]. Increases in IL-1β, TNFα, and IL-6 are accompanied by elevations in IDO levels in the whole brains of mice 4 h after a single icv Tat injection, followed by increased depressive-like behavior 24 h after Tat exposure [61]. Increased levels of IDO are found in PWH and associated with increased neuropsychiatric symptoms [150] and inflammation despite viral suppression with ART [151]. In a similar Tat tg model with multiple *tat* gene copy numbers and more aggressive pathology, increased striatal serotonin and 5-HIAA levels are associated with increased reward response thresholds in an intracranial self-stimulation paradigm after 3 days of Tat exposure [93]. However, serotonin and 5-HIAA decreased below control levels after 40 days of Tat exposure [93]. Together, these data suggest that HIV-induced depression is associated with decreased serotonin levels resulting from increased IDO activity caused by Tat-induced increases in proinflammatory cytokines.

Although we previously saw no effect of Tat on ACC layer 5 pyramidal neuron dendritic spine density, branching, or length [59], in the present study, Tat uniformly decreased the density of dendritic spines on layer 5 pyramidal neurons in the ACC. By contrast, we found no evidence of dendritic degeneration, although Tat has been shown to induce dendritic damage in PFC neurons in vitro [152,153,154]. These disparate results likely reflect the inherent vulnerability of neurons to toxic insults in vitro. We previously found that Tat decreases inhibitory synaptic connections within the ACC region of the prefrontal cortex [59], which may precede alterations in spine density. We also saw no changes in dendritic spine density with morphine alone or in combination with Tat. This differs from what we see in the striatum [47] and highlights the regional differences in Tat- and opioid-induced neuronal pathology. Others have shown that two or four weeks of morphine administration decreases PFC pyramidal neuronal density in rats one month after the last morphine dose [155,156], suggesting that more prolonged morphine administration might increase dendritic pathology within the PFC. Alternatively, differences in Tat- and opioid-induced synaptodendritic damage between the present and previous studies may be due to rapid changes in synaptic plasticity and dendritic spine stability. We previously found that within the stratum lacunosum-moleculare, but not other layers within the hippocampal area CA1, Tat and morphine withdrawal interacted to decrease the density of thin/filopodial spines in pyramidal neurons, but not other morphological spine subtypes considered more mature [157]. Future studies investigating longer-term depression or potentiation in response to Tat and opioids to measure alterations in synaptic plasticity in the PFC are warranted.

## 5. Conclusions

PWH and OUD are at increased risk for depression and a decreased quality of life. The present study extends previous findings by demonstrating that the HIV protein Tat and the opioid agonist morphine induce anhedonia and decrease well-being and adaptability that are accompanied by neuroimmune and morphological alterations in the PFC. The increased depressive-like behavior coincided with Tat-induced increased levels of the proinflammatory chemokine CCL3 in the PFC and decreased dendritic spine density on pyramidal neurons within layer 5 of the ACC. By contrast, within the PFC, morphine augmented Tat-induced increases in the anti-inflammatory cytokine IL-10, and Tat tended to decrease proinflammatory IL-6 and IFN-γ levels, further supporting previous research suggesting that sustained exposure to Tat and morphine eventually fatigues the innate immune response throughout multiple brain regions that mediate emotionality. Across all treatment groups, increased depressive-like behavior and decreased mobility correlated with increases in proinflammatory IL-6, CXCL1, and CCL11 and anti-inflammatory IL-10. Depressive-like behaviors are accompanied by morphological changes in ACC output layer pyramidal neurons that appear to be mediated by preceding inflammatory cytokine expression within the PFC despite evidence of neuroimmune suppression following sustained Tat insult.

## Figures and Tables

**Figure 1 viruses-15-00590-f001:**
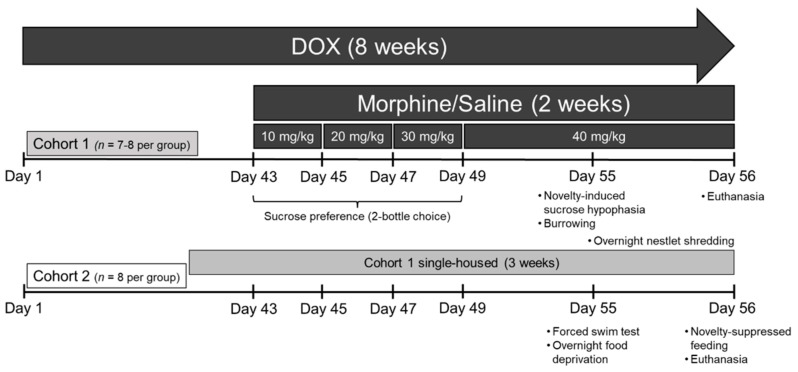
In vivo experimental timeline. Tat(+) and Tat(–) (control) mice were divided into two behavioral cohorts and administered doxycycline (DOX)-containing chow throughout the entire experiment (8 weeks) to induce Tat expression (or control for off-target effects). During the last two weeks of DOX administration, all mice were administered ramping morphine (10–40 mg/kg, increasing by 10 mg/kg/2 day, s.c., b.i.d.) or saline. Cohort 1 mice were single-housed for 1 week to acclimate before morphine administration and remained single-housed till the end of the study. Cohort 1 mice were assessed for sucrose preference in the two-bottle choice assay during the first week of morphine administration. Mice were then allowed to abstain from sucrose for one week, with ad libitum food access, before being assessed for anhedonia in the novelty-induced sucrose hypophagia assay and for innate well-being behaviors (including the burrowing and nestlet shredding assays). On day 13 of repeated morphine (or saline) administration, cohort 2 mice were assessed for acute stressor coping behavior in the forced swim test, food-deprived overnight, and assessed for depressive-like behavior in the novelty-suppressed feeding test the following day. Immediately after the conclusion of the nestlet shredding or novelty-suppressed feeding assays, brain tissues were randomly assigned to groups for assessment by (i) multiplex cytokine assay to determine cytokine/chemokine levels or (ii) morphologic analysis using a Rapid Golgi impregnation procedure.

**Figure 2 viruses-15-00590-f002:**
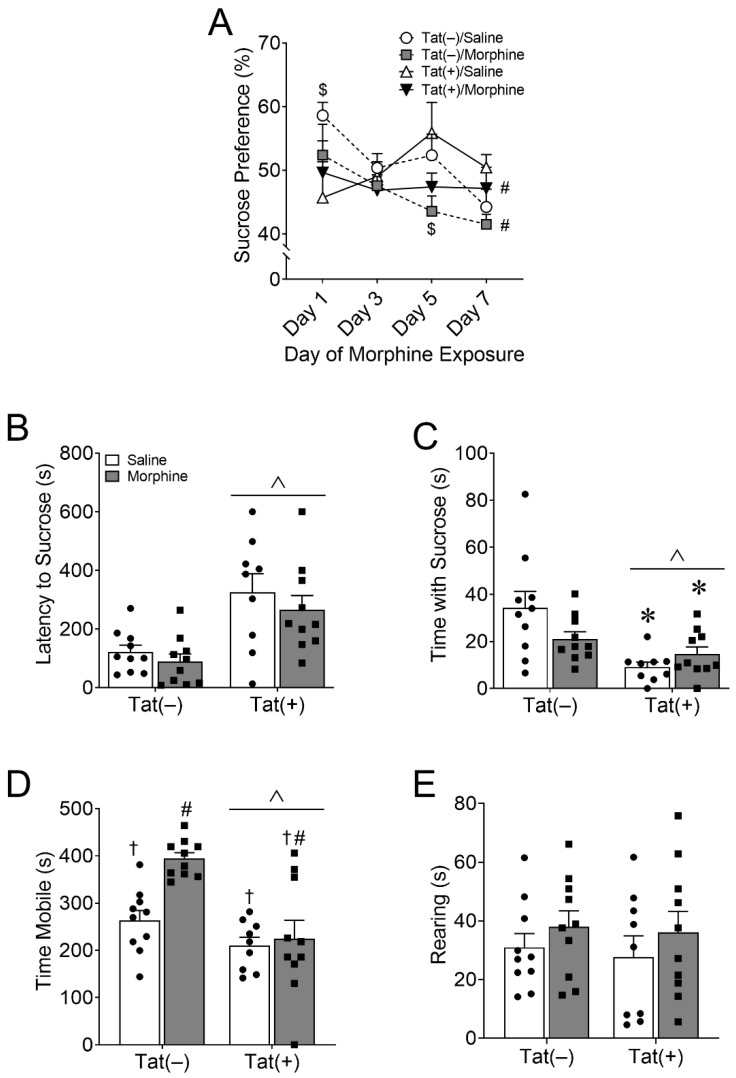
Morphine decreased sucrose preference, whereas HIV-1 Tat decreased novelty-induced sucrose hypophagia. Escalating doses of morphine (10–40 mg/kg, increasing by 10 mg/kg/2 day, s.c., b.i.d.) for 1 week decreased home cage sucrose preference in the two-bottle choice test. In the two-bottle choice test, home cage sucrose preference was measured every other day (i.e., 48 h) in both Tat(–) and Tat(+) mice (**A**). Tat(+) mice that received saline exhibited decreased sucrose preference on the first day of home cage sucrose administration that was reversed on day 5 (**A**). Tat(+) mice also exhibited an increased latency to seek sucrose (**B**) and decreased interactions (**C**) with sucrose in a novel environment. Morphine exposure increased, whereas Tat exposure decreased locomotor activity (**D**). Planned comparisons indicated that Tat(–) morphine-treated mice were more mobile than mice in all other treatment groups (**D**). There were no differences in rearing behavior (**E**). Data are presented as the mean ± the SEM; *n* = 9–10 mice per group; ^$^ *p* < 0.05 interaction vs. Tat(+) Saline; * *p* < 0.05 interaction vs. Tat(–) Saline; ^†^ *p* < 0.05 interaction vs. Tat(–) Morphine; ^ *p* < 0.05 main effect of Tat; ^#^ *p* < 0.05 main effect of morphine.

**Figure 3 viruses-15-00590-f003:**
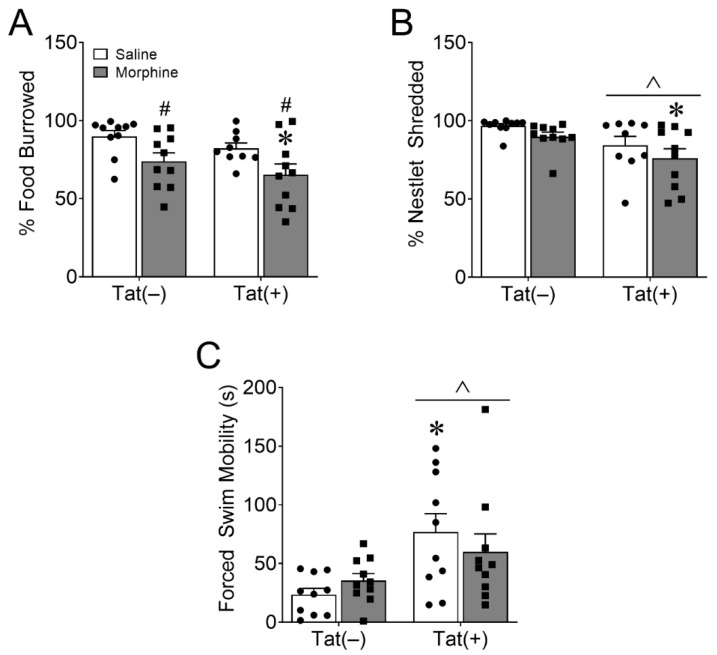
Morphine decreased burrowing, whereas HIV-1 Tat decreased nestlet shredding and adaptability. The main effects indicate that escalating morphine exposure (10–40 mg/kg, increasing by 10 mg/kg/2 day, s.c., b.i.d.) decreased burrowing of food pellets (**A**), whereas Tat exposure for eight weeks decreased shredding of a pressed cotton nestlet (**B**). However, planned comparisons indicate that Tat(+) and morphine co-exposure increased burrowing (**A**) and nestlet shredding (**B**) significantly more than Tat(–) and saline-treated mice, suggesting that Tat(+) and morphine co-exposure are driving the main effects. Tat exposure also increased overall mobility in the forced swim test. Planned comparisons indicate that saline-treated Tat(+) mice were significantly more mobile than saline-exposed Tat(–) mice (**C**). Data are presented as the mean ± the SEM; *n* = 9–10 mice per group; * *p* < 0.05 interaction vs. Tat(–) Saline; ^ *p* < 0.05 main effect of Tat; ^#^ *p* < 0.05 main effect of morphine.

**Figure 4 viruses-15-00590-f004:**
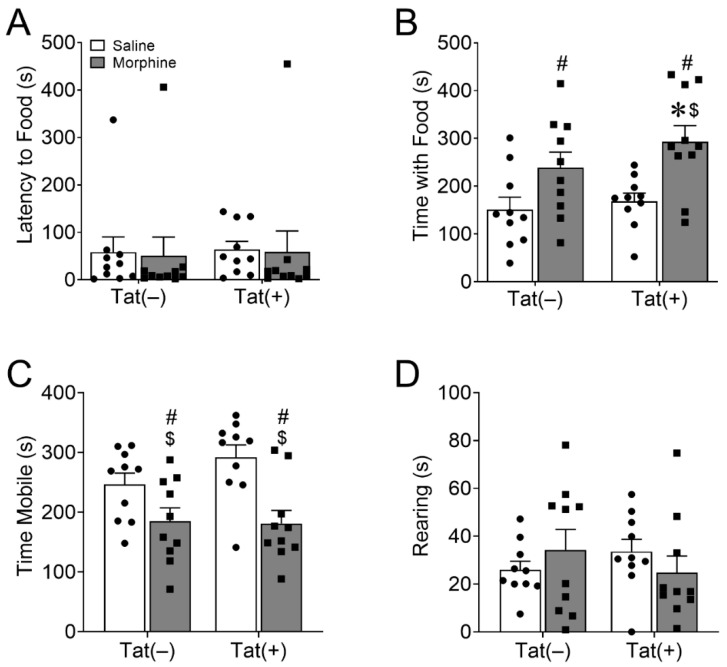
Morphine, but not HIV-1 Tat, increased feeding in a novel environment after food deprivation. The main effects indicated that ramping morphine (10–40 mg/kg, increasing by 10 mg/kg/2 day, s.c., b.i.d.), regardless of genotype, increased doxycycline pellet feeding time in a novel environment after overnight food deprivation (**B**) with no difference in latency (**A**). Planned comparisons revealed that Tat(+) morphine-exposed mice spent more time feeding than Tat(–) or Tat(+) and saline co-exposed mice (**B**). Morphine-treated mice also spent less time mobile in the novelty-suppressed feeding test (**C**), but there was no effect of morphine or Tat on rearing behavior (**D**). Data are presented as the mean ± the SEM; *n* = 10 mice per group; ^$^ *p* < 0.05 interaction vs. Tat(+) Saline; * *p* < 0.05 interaction vs. Tat(–) Saline; ^#^
*p* < 0.05 main effect of morphine.

**Figure 5 viruses-15-00590-f005:**
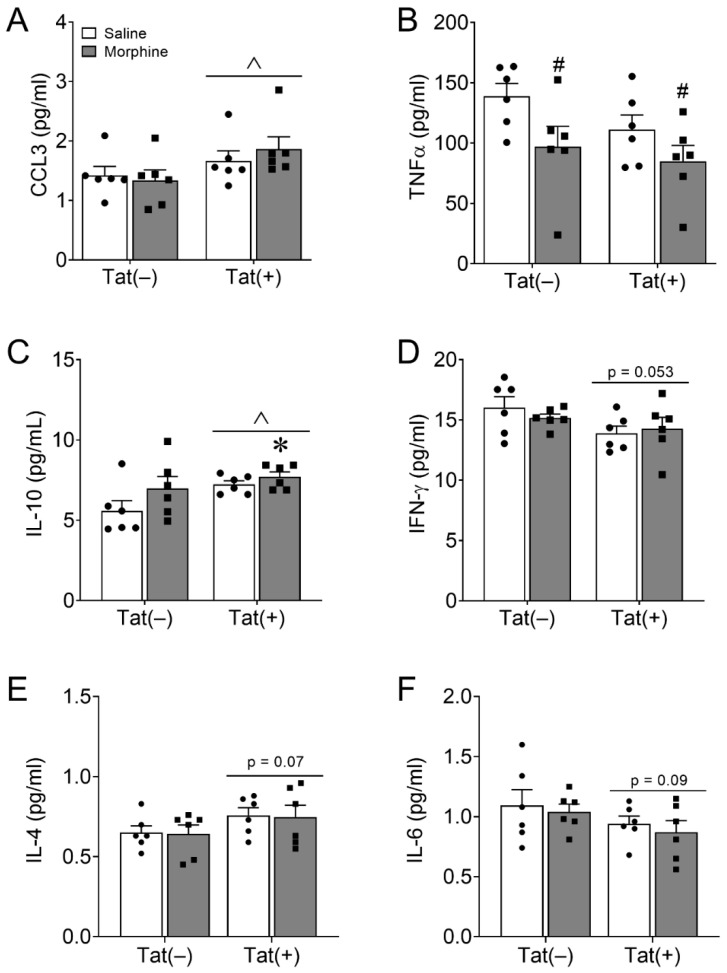
HIV-1 Tat increases CCL3 and IL-10, whereas morphine decreases TNFα expression in the prefrontal cortex (PFC). Multiplex analyses were performed to assess levels of 23 cytokines within the PFC of Tat(−) and Tat(+) mice co-administered saline or morphine (Table 1). Eight weeks of Tat exposure increased levels of the proinflammatory chemokine CCL3 (**A**) and anti-inflammatory cytokine IL-10 (**C**). Planned comparisons revealed that Tat(+) and morphine co-exposed mice had increased IL-10 PFC levels compared to Tat(–) and saline co-exposed mice (**C**). Two weeks of morphine administration decreased PFC levels of the proinflammatory cytokine TNFα irrespective of Tat exposure (**B**). Tat exposure also tended to decrease levels of the proinflammatory IFN-γ (**D**) and IL-6 (**F**) while tending to increase levels of the anti-inflammatory IL-4 (**E**). Data are presented as the mean ± the SEM; *n* = 6 mice per group; non-significant chemokines and cytokines assessed are presented in Table 2. * *p* < 0.05 interaction vs. Tat(–) Saline; ^ *p* < 0.05 main effect of Tat; ^#^
*p* < 0.05 main effect of morphine.

**Figure 6 viruses-15-00590-f006:**
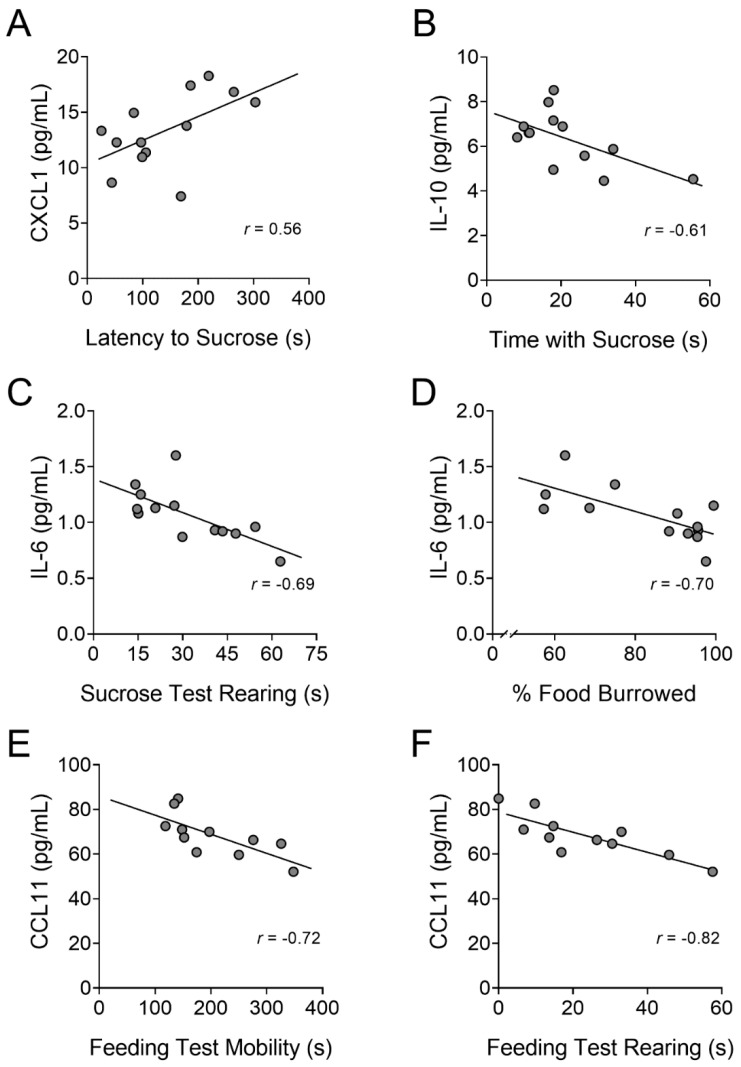
Increased PFC levels of CXCL1, CCL11, IL-6, and IL-10 positively correlated with depressive-like behavior and hypolocomotion. There was a direct correlation between increased levels of CXCL1 in the PFC and an increased latency to seek sucrose in the novelty-induced hypophagia test (**A**), whereas increased IL-10 expression inversely correlated with sucrose consumption (**B**). Increased levels of the proinflammatory cytokine IL-6 in the PFC negatively correlated with the time spent rearing in the novelty-induced hypophagia test (**C**) and the percentage of food burrowed in the burrowing test of well-being (**D**). There was also an inverse correlation between CCL11 and mobility (**E**) and rearing (**F**) in the novelty-suppressed feeding test. Correlation coefficient *z*-tests were performed with hypothesized correlations equal to 0. After false discovery rate (FDR) corrections for performing multiple *z* tests were applied, only the inverse correlation between CCL11 PFC levels and rearing behavior in the novelty-suppressed feeding test was significant; *n* = 11–13 mice per group.

**Figure 7 viruses-15-00590-f007:**
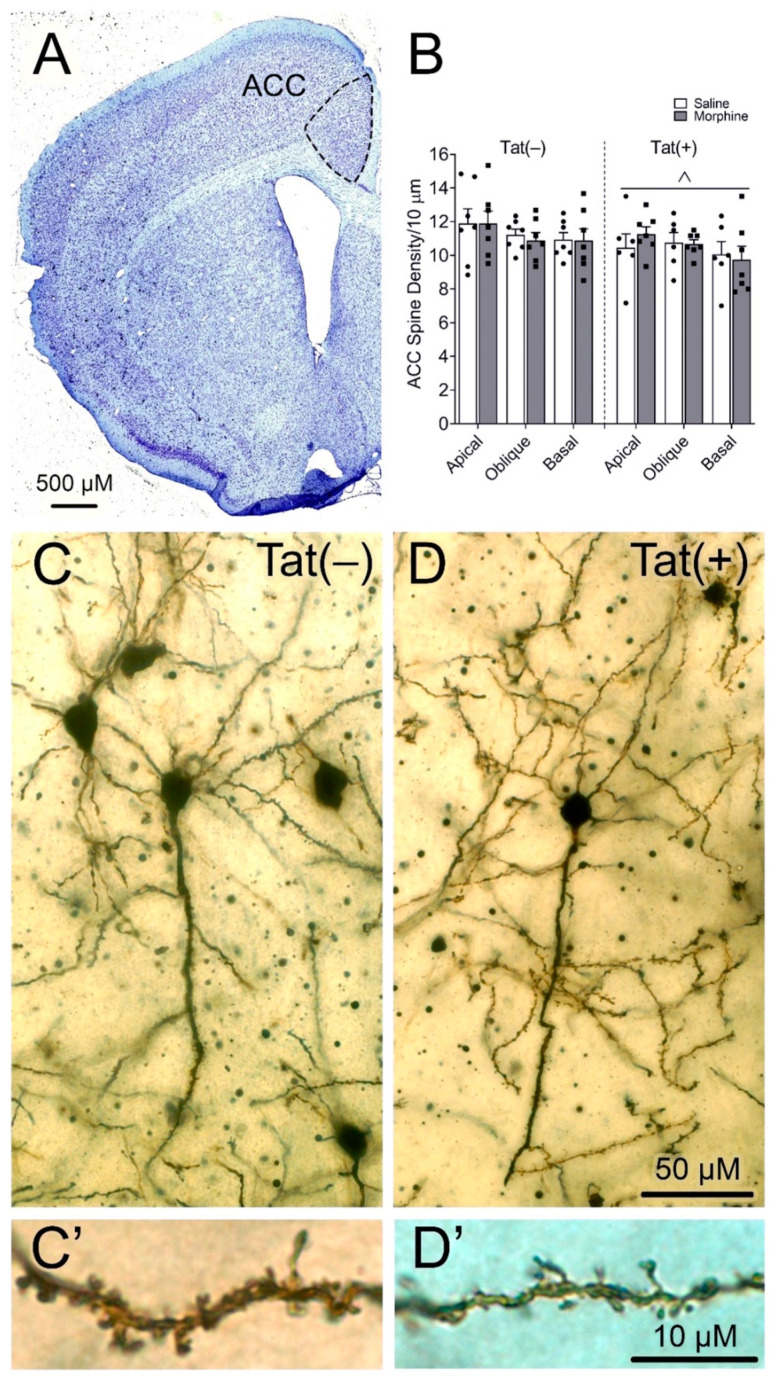
HIV-1 Tat, but not morphine, decreases layer V pyramidal neuron dendritic spine density within the anterior cingulate cortex (ACC). Representative Nissl-stained image of the ACC region of the prefrontal cortex (PFC) (scale bar = 500 µM) (**A**). Tat decreased dendritic spine density across apical, oblique, and basal dendrites irrespective of morphine treatment (**B**). Representative lower magnification images of Golgi-impregnated layer V pyramidal neurons (scale bar = 50 µM) in saline-treated Tat(–) (**C**) and Tat(+) (**D**) mice. Higher magnification images illustrating dendritic spines in saline-treated Tat(–) (**C’**) and Tat(+) (**D’**) mice (scale bar = 10 µM). Data are presented as the mean ± the SEM; multiple neurons (≥6) were sampled from each mouse; *n* = 6–7 mice per group; ^^^ *p* < 0.05, the main effect of Tat.

**Table 1 viruses-15-00590-t001:** Effects of Tat and morphine on prefrontal cortical chemokine and cytokine levels.

Cytokine	Main Effect of Tat		Main Effect of Morphine		Interaction	
	*F* _(1,20)_	*p*	*F* _(1,20)_	*p*	*F* _(1,20)_	*p*
CCL2	0.84	0.37	2.36	0.14	0.27	0.61
CCL3	4.82	**0.04 ^**	0.11	0.74	0.67	0.42
CCL4	0.96	0.34	0.11	0.75	0.23	0.64
CCL5	0.07	0.79	0.26	0.62	0.37	0.55
CCL11	0.13	0.72	0.52	0.48	0.91	0.35
CXCL1	2.04	0.17	0.94	0.34	0.05	0.82
G-CSF	0.02	0.89	0.44	0.51	0.35	0.56
GM-CSF	0.01	0.96	0.50	0.49	0.10	0.76
TNFα	2.19	0.15	6.49	**0.02 ^#^**	0.35	0.56
IFN-γ	4.2	**0.06**	0.10	0.75	0.74	0.40
IL-1α	1.21	0.28	0.42	0.53	0.23	0.64
IL-1β	0.26	0.61	0.03	0.86	1.03	0.72
IL-2	0.01	0.92	0.07	0.80	0.34	0.56
IL-3	1.34	0.25	1.79	0.20	0.01	0.94
IL-6	3.00	**0.09**	0.44	0.52	0.01	0.92
IL-9	<0.01	0.95	0.57	0.46	1.02	0.33
IL-12 (p40)	0.92	0.35	0.75	0.40	0.18	0.67
IL-12 (p70)	2.34	0.14	0.19	0.67	0.40	0.53
IL-17A	0.04	0.84	0.75	0.40	0.18	0.68
IL-4	3.61	**0.07**	0.03	0.87	<0.01	0.99
IL-5	0.69	0.42	0.95	0.34	0.02	0.89
IL-10	5.09	**0.04 ^**	3.23	**0.09**	0.78	0.39
IL-13	0.28	0.60	0.01	0.90	0.21	0.65

The F-values, degrees of freedom, and *p*-values from the two-way ANOVA results of the 23 cytokines and chemokines assayed via Bio-Rad multiplex analyses. Bolded values denote significant *p* < 0.05 or trending *p* < 0.10 *p*-values. ^ *p* < 0.05 denotes a Tat main effect; ^#^ *p* < 0.05 denotes a morphine main effect.

**Table 2 viruses-15-00590-t002:** Means ± SEM of chemokines and cytokines not significantly altered by Tat or morphine in this study.

	Tat(–) Saline		Tat(–) Morphine		Tat(+) Saline		Tat(+) Morphine	
Cytokine (pg/mL)	Mean	SEM	Mean	SEM	Mean	SEM	Mean	SEM
CCL2	9.37	1.83	13.52	3.74	10.98	1.38	19.41	6.92
CCL4	35.83	1.81	36.98	1.73	37.91	1.10	37.70	0.87
CCL5	12.59	0.55	13.41	0.74	13.23	0.70	13.16	0.89
CCL11	59.92	4.30	67.46	4.65	65.85	4.51	64.81	4.50
CXCL1	11.61	1.30	13.12	1.66	13.70	0.72	14.63	1.18
G-CSF	1.42	0.36	1.79	0.35	1.64	0.20	1.66	0.23
GM-CSF	1.67	0.93	2.50	1.20	1.88	0.39	2.20	0.46
IL-1α	4.03	0.18	4.07	0.20	4.16	0.22	4.40	0.25
IL-1β	11.68	0.98	12.41	1.37	12.87	1.48	12.61	1.56
IL-2	20.73	1.55	22.24	1.81	21.97	1.89	21.38	1.89
IL-3	0.44	0.10	0.58	0.10	0.56	0.03	0.68	0.12
IL-9	22.42	2.01	18.53	1.09	20.06	1.12	20.62	3.61
IL-12 (p40)	7.22	1.16	7.90	1.25	8.05	1.43	10.07	2.19
IL-12 (p70)	101.19	13.61	87.65	7.30	73.86	11.32	76.34	16.51
IL-17A	6.41	1.24	7.78	1.52	6.65	0.45	7.12	0.68
IL-5	0.72	0.08	0.78	0.13	0.77	0.04	0.86	0.04
IL-13	49.44	3.27	52.06	5.03	53.92	4.10	52.40	5.43

The means and standard errors of the means (SEM) expressed in pg/mL for prefrontal cortex (PFC) chemokine and cytokine protein levels in Tat(+) and Tat(–) mice that were not significantly changed by eight weeks Tat or two weeks of morphine exposure.

## Data Availability

The data presented in this study are available on request from the corresponding author.

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
