# Peer review of "Depressive-like Behavior Is Accompanied by Prefrontal Cortical Innate Immune Fatigue and Dendritic Spine Losses after HIV-1 Tat and Morphine Exposure"

_viruses, 2023, doi:10.3390/v15030590_

Round 1
Reviewer 1 Report
Manuscript entitled “Depressive-like behavior is accompanied by prefrontal cortical innate immune fatigue and dendritic spine losses after HIV-1 Tat and morphine exposure” I found very interesting and novel in which authors discuss about morphine addiction in patients with HIV and how it will affect the motivational system. I will recommend this manuscript for publication after minor revision.
I must appreciate the designed experimental plan and rationally performed all behavioral experiments. Figure 3A: Tat(+) morphine group needs minor correction. Authors did a unique analysis in figure 6 “correlation of chemokines and cytokines with depressive behavior”.
Overall authors gave nice justification for all the experiments performed in the present manuscript. However, I still have some questions what was the rationale behind doing molecular analysis from mPFC only? Authors could use more specific more specific brain regions which are primarily involved in disease pathophysiology.
It would be great if authors could include one more paragraph in discussion which includes how chemokines and cytokines affect the depression associated monoamines?
Reviewer 2 Report
This manuscript reports the effects of tat and morphine exposure on depressive-like behaviors and prefrontal cortex cytokine/chemokine levels and spine density. The study is strengthened by the use of multiple measures of depressive-like behavior and makes a meaningful contribution to the field of cognitive and emotional effects of HIV. Comments are listed below:
- It is unclear if the novelty-induced hypophagia and novelty suppressed feeding tests are different from one another. There is no consistency in reporting the testing duration, food access, or timing of morphine exposure between the two tests.
- The method of euthanasia should be stated.
- L248: The statement “No values were above detection limits” is not consistent with the data that are presented in Table 1.
-
- Table 2 should include units of measure
- Fig 2A: It is unclear if each day that is reported represents 24 h or 48 h. If it is 24 h, the figure is only showing sucrose preference from sucrose being on one side of the cage. The figure should include all days and include bottle side as a covariate if necessary.
- Fig 2: the use of multiple symbols is confusing. For example, # is stated to represent a significant main effect of morphine, but is used over one saline and one morphine bar.
- There is no justification for the use of Tukey’s post hoc tests vs. planned comparisons in different analyses.
- It appears that multiple correlational analyses were performed without controlling for false discovery. Considering the large number of cytokines/chemokines measured, an approach such as principal components regression may be more appropriate.
- Were any tests for correlation between cytokines/chemokines and spine density?
- Discussion: “cogent” appears to be used in place of “cognate”
- Line 585: should “proceed” be replaced with “precede”?
- Of note, there appear to be different font sizes used throughout the manuscript
-
Reviewer 3 Report
Depressive-like behavior is accompanied by prefrontal cortical innate immune fatigue and dendritic spine losses after HIV-1 Tat and morphine exposure
This is an excellent original research article addressing the comorbid condition of HIV and opioid abuse causing depression-like behavior as observed in the prefrontal cortex (PFC) innate immune fatigue and dendritic spine losses. The study was conducted on male mice to simulate the condition of people living with HIV (PWH) and those who use opioids. The study concluded that HIV-1 Tat and morphine differentially induce depressive-like behaviors associated with increased neuroinflammation, synaptic losses, and immune fatigue within the PFC. I found the study well-designed and meticulously executed, the manuscript is well-written. Following are the specific comments to further strengthen the present manuscript,
1. The introduction can be summarized.
2. In figure 1. Control 1 and 2, also try to provide a line for Tat.
3. What if the study was conducted longer, which is the near-real conditions of PWH?
Round 2
Reviewer 2 Report
Accept.